# Scalable Bayesian dynamic covariance modeling with variational Wishart and inverse Wishart processes

**Creighton Heaukulani**
No Affiliation
Bangkok, Thailand
c.k.heaukulani@gmail.com

**Mark van der Wilk**
PROWLER.io
Cambridge, United Kingdom
mark@prowler.io

## Abstract

We implement gradient-based variational inference routines for Wishart and inverse Wishart processes, which we apply as Bayesian models for the dynamic, heteroskedastic covariance matrix of a multivariate time series. The Wishart and inverse Wishart processes are constructed from i.i.d. Gaussian processes, existing variational inference algorithms for which form the basis of our approach. These methods are easy to implement as a black-box and scale favorably with the length of the time series, however, they fail in the case of the Wishart process, an issue we resolve with a simple modification into an additive white noise parameterization of the model. This modification is also key to implementing a factored variant of the construction, allowing inference to additionally scale to high-dimensional covariance matrices. Through experimentation, we demonstrate that some (but not all) model variants outperform multivariate GARCH when forecasting the covariances of returns on financial instruments.

## 1 Introduction

Estimating the (time series of) covariance matrices between the variables in a multivariate time series is a principal problem of interest in many domains, including the construction of financial trading portfolios [15] and the study of brain activity measurements in neurological studies [7]. Estimating the entries of the covariance matrices is a challenging problem, however, because there are $O(ND^2)$ parameters to estimate for a time series of length $N$ with $D$ variables, yet we only record a *single* observation of the time series consisting of $O(ND)$ data points. Bayesian models (and their corresponding inference procedures) often perform well in these overparameterized problems; indeed, Fox and West [7], Wilson and Ghahramani [24] and Fox and Dunson [6] show that Bayesian approaches based on the *Wishart* and *inverse Wishart processes* produce better estimates of dynamic covariance matrices than the venerable multivariate GARCH approaches [3, 4].

The Wishart and inverse Wishart processes are two related stochastic processes in the state space of symmetric, positive definite matrices, making them appropriate models for (heteroskedastic) time series of covariance matrices. They are themselves constructed from i.i.d. Gaussian processes, in analogy to the construction of Wishart or inverse Wishart random variables from i.i.d. Gaussian random variables. Exact posterior inference for these models is intractable, and so previous authors have suggested approximate inference routines based on Markov chain Monte Carlo (MCMC) algorithms. We instead propose a gradient-based variational inference routine, derived from approaches to approximate inference with (sparse and/or multi-output) Gaussian process models. Taking the variational approach has several advantages including a simple, black-box implementation and the ability to scale down the computational cost of inference with respect to $N$, the length of the time series, if required. Furthermore, we derive a *factored* variant of the model that may additionally scale inference to large numbers of variables $D$, i.e., the dimensionality of the covariance matrix. In our

experiments, we will see that our black-box, scalable, gradient-based variational inference routines have predictive performance that is competitive with multivariate GARCH.

We start by considering variational inference routines for the model presented by Wilson and Ghahramani [24]; our approach is a gradient-based analogue of the coordinate ascent algorithms for variational inference on Wishart processes presented by van der Wilk et al. [23]. The factored variants of the model that we build toward in Section 5 end up being a reparameterization of the construction by Fox and Dunson [6], and so our work provides gradient-based variational inference routines for their model class as well. Alternatively, Fox and West [7] construct inverse Wishart processes that are autoregressive (as opposed to the full process dependence assumed by the Gaussian processes); Wu et al. [26] model time series of (univariate) variances with Gaussian process models; and Wu et al. [25] consider generalizing multivariate GARCH by modeling the transition matrices of the process with autoregressive structures. All of these references elect MCMC-based inference and emphasize that Bayesian inference of (co)variances dominate non-Bayesian approaches.

## 2   Wishart and inverse Wishart processes

Let $Y := (Y_n, \ n \geq 1)$ denote a sequence of measurements in $\mathbb{R}^D$, which will be regressed upon a corresponding sequence of input locations (i.e., *covariates*) in $\mathbb{R}^p$ denoted by $X := (X_n, \ n \geq 1)$. In our applications, we will take $X_n$ to be a univariate (so $p = 1$), real-valued representation of the "time" at which the measurement $Y_n$ was taken. For example, in a dataset of daily stock returns, the vector $Y_n$ can record the returns for $D$ stocks on day $n$, for $n \leq N$, where the points in $X$ can be linearly spaced in some fixed interval like $(0, 1)$, and individual points of $X$ may be altered to account for any irregular spacing, such as weekends or the removal of special trading days.

We let the conditional likelihood of $Y_n$ be given by the multivariate Gaussian density

$$Y_n \mid \mu_n, \Sigma_n \ \sim \ \mathcal{N}(\mu_n, \Sigma_n), \qquad n \geq 1, \tag{1}$$

for a sequence $\mu_1, \mu_2, \ldots$ of elements in $\mathbb{R}^D$ and a sequence $\Sigma_1, \Sigma_2, \ldots$ of (random) positive definite matrices, which we note may depend on $X$. In modern portfolio theory [15], where $Y_n$ is a sequence of financial returns, predictions for the mean process $\mu_1, \mu_2, \ldots$ are used in conjunction with predictions for the *covariances of the residuals* $\Sigma_1, \Sigma_2, \ldots$ to construct a portfolio that maximizes expected return while minimizing risk. In this article, we will focus on modeling the process $\Sigma_1, \Sigma_2, \ldots$, and we henceforth assume that $Y_n$ is mean zero (i.e., $\mu_n = 0$, the zero vector), for $n \leq N$.

Bayesian models for the sequence $\Sigma := (\Sigma_1, \Sigma_2, \ldots)$ include the Wishart and inverse Wishart processes. In analogy to the construction of Wishart and inverse Wishart random variables from i.i.d. collections of Gaussian random variables, we may construct Wishart and inverse Wishart processes from i.i.d. collections of Gaussian processes as follows. Let

$$f_{d,k} \ \sim \ \mathrm{GP}(0, \kappa(\,\cdot\,, \,\cdot\,; \theta)), \qquad d \leq D, \ k \leq \nu, \tag{2}$$

be i.i.d. Gaussian processes with zero mean function and (shared) kernel function $\kappa(\,\cdot\,, \,\cdot\,; \theta)$, where $\theta$ denotes any parameters of the kernel function, and the positive integer-valued $\nu \geq D$ will be called the *degrees of freedom* parameter. Let $F_{n,d,k} := f_{d,k}(X_n)$, and let $F_n := (F_{n,d,k}, d \leq D, \ k \leq \nu)$ denote the $D \times \nu$ matrix of collected function values, for every $n \geq 1$. Construct

$$\Sigma_n = A F_n F_n^T A^T, \qquad n \geq 1, \tag{3}$$

where $A \in \mathbb{R}^{D \times D}$ satisfies the condition that the symmetric matrix $AA^T$ is positive definite.[1] So constructed, $\Sigma_n$ is (marginally) Wishart distributed, and $\Sigma := (\Sigma_1, \Sigma_2, \ldots)$ is correspondingly called a *Wishart process with degrees of freedom $\nu$ and scale matrix $AA^T$*. Alternatively, if we instead construct the precision matrix

$$\Sigma_n^{-1} = A F_n F_n^T A^T, \qquad n \geq 1, \tag{4}$$

then $\Sigma_n$ is inverse Wishart distributed, and $\Sigma$ is called an *inverse Wishart process (with degrees of freedom $\nu$ and scale matrix $AA^T$)*. The dynamics of the process of covariance matrices $\Sigma$ are inherited by the Gaussian processes, which are perhaps best controlled by the kernel function $\kappa(\cdot, \cdot; \theta)$.

The posterior distribution for $\Sigma$ is difficult to evaluate, and so previous MCMC-based approaches to approximate inference typically utilize conjugacy results between the (inverse) Wishart distribution and the likelihood function in Eq. (1). In contrast, the "black-box" variational inference routines that we suggest only require evaluations of the log conditional likelihood function, dramatically simplifying their implementation. For the Wishart process case, we have

$$\log p(Y_n \mid F_n) = -\frac{D}{2}\log(2\pi) - \frac{1}{2}\log|AF_nF_n^TA^T| - \frac{1}{2}Y_n^T(AF_nF_n^TA^T)^{-1}Y_n, \quad (5)$$

and for the inverse Wishart case, we have

$$\log p(Y_n \mid F_n) = -\frac{D}{2}\log(2\pi) + \frac{1}{2}\log|AF_nF_n^TA^T| - \frac{1}{2}Y_n^TAF_nF_n^TA^TY_n. \quad (6)$$

Changing our implementation between these two likelihood models only requires changing the line(s) of code computing these expressions, highlighting the ease of the black-box approach. Other likelihoods may be considered; for example, Eq. (5) and Eq. (6) may be replaced by the likelihood function for a *multivariate t-distribution* (as done by Wu et al. [25]), a popular heavy-tailed model.

## 3   Inducing points and variational inference

A popular approach to variational inference with Gaussian processes is based on the introduction of $M$ *inducing points* $Z := (Z_1, \ldots, Z_M)$, taking values in the same space as the inputs $X$, upon which we assume the dependence of the function values $F_n$ decouple during inference [1, 9, 18]. In particular, for every $d \leq D$ and $k \leq \nu$, let $U_{m,d,k} := f_{d,k}(Z_m)$, for $m \leq M$, denote the evaluations of the Gaussian process at the inducing points, and collectively denote $U_{d,k} := (U_{m,d,k}, m \leq M)$ and $F_{d,k} := (F_{n,d,k}, n \leq N)$. By independence, and with well-known properties of the Gaussian distribution, we may write

$$p(Y, F, U) = \prod_{n=1}^{N}\Big[p(Y_n \mid F_n)\Big]\prod_{d=1}^{D}\prod_{k=1}^{\nu}\Big[p(F_{d,k} \mid U_{d,k})p(U_{d,k})\Big], \quad (7)$$

where

$$p(F_{d,k} \mid U_{d,k}) = \mathcal{N}(F_{d,k}; K_{xz}K_{zz}^{-1}U_{d,k}, K_{xx} - K_{xz}K_{zz}^{-1}K_{xz}^T), \quad (8)$$
$$p(U_{d,k}) = \mathcal{N}(U_{d,k}; 0, K_{zz}), \quad (9)$$

and where the $N \times N$ matrix $K_{xx}$ has $(n, n')$-th element $\kappa(X_n, X_{n'}; \theta)$, the $N \times M$ matrix $K_{xz}$ has $(n, m)$-th element $\kappa(X_n, Z_m; \theta)$, and the $M \times M$ matrix $K_{zz}$ has $(m, m')$-th element $\kappa(Z_m, Z_{m'}; \theta)$.

Following Hensman et al. [10], we introduce a variational approximation to the posterior distribution of the latent variables that takes the following form: Independently for every $d \leq D$ and $k \leq \nu$, let

$$q(F_{d,k}, U_{d,k}) = p(F_{d,k} \mid U_{d,k})q(U_{d,k}), \quad \text{where } q(U_{d,k}) = \mathcal{N}(U_{d,k}; \mu_{d,k}, S_{d,k}), \quad (10)$$

for some *variational parameters* $\mu_{d,k} \in \mathbb{R}^M$ and $S_{d,k} \in \mathbb{R}^{M \times M}$ a real, symmetric, positive definite matrix. It follows that

$$q(F_{d,k}) = \int p(F_{d,k} \mid U_{d,k})q(U_{d,k})\mathrm{d}U_{d,k} = \mathcal{N}(F_{d,k}; \tilde{K}\mu_{d,k}, K_{xx} + \tilde{K}(S_{d,k} - K_{zz})\tilde{K}^T), \quad (11)$$

where $\tilde{K} := K_{xz}K_{zz}^{-1}$. That is, the variational approximation $q(U_{d,k})$ *induces* the approximation $q(F_{d,k})$. We may then lower bound the log marginal likelihood of the data as follows:

$$\log p(Y) \geq \sum_{n=1}^{N}\mathbb{E}_{q(F_n)}[\log p(Y_n \mid F_n)] - \sum_{d=1}^{D}\sum_{k=1}^{\nu}\mathrm{KL}[q(U_{d,k}) \,\|\, p(U_{d,k})], \quad (12)$$

where $\mathrm{KL}[q \,\|\, p]$ denotes the Kullback–Leibler divergence from $q$ to $p$. To perform inference, we maximize the *evidence lower bound* on the right hand side of Eq. (12)—which we note depends on the parameters to be optimized $\Theta := \{Z, \mu, S, \theta\}$ through the variational distribution $q$—via gradient ascent. The terms $\mathrm{KL}[q(U_{d,k}) \,\|\, p(U_{d,k})]$ may be analytically evaluated and so their gradients (w.r.t. the optimization parameters) are straightforward to compute. Because $q(F)$ is not conjugate to the likelihood $p(Y \mid F)$, we cannot analytically evaluate the term $\mathbb{E}_{q(F_n)}[\log p(Y_n \mid F_n)]$. We therefore

follow Salimans and Knowles [21] and Kingma and Welling [12] to approximate the gradients of (Monte Carlo estimates of) this expression by "differentiating through" random samples from $q$ as follows. Independently for every $d \le D$ and $k \le \nu$, produce the $R \ge 1$ Monte Carlo samples

$$F_{d,k}^{(r)} = \psi_{d,k}(w_{d,k}^{(r)}; \Theta), \quad w_{d,k}^{(r)} \sim \mathcal{N}(0, I_N), \qquad r = 1, \ldots, R, \tag{13}$$

where $\psi_{d,k}(w; \Theta) := B_{d,k}w + \tilde{K}\mu_{d,k}$, for the matrix $B_{d,k} \in \mathbb{R}^{N \times N}$ that satisfies $B_{d,k}B_{d,k}^T = K_{xx} + \tilde{K}(S_{d,k} - K_{zz})\tilde{K}^T$, as given by the Cholesky factor. Note then that the samples generated according to Eq. (13) have distribution $q(F_{d,k})$. Form the Monte Carlo approximations

$$\nabla_{(\mu_{d,k}, S_{d,k})} \mathbb{E}_{q(F_n)}[\log p(Y_n \mid F_n)] \approx \frac{1}{R} \sum_{r=1}^{R} \Big[ \nabla_{F_{d,k}} \log p(Y_n \mid F_n^{(r)}) \circ \nabla_{(\mu_{d,k}, S_{d,k})} \psi_{d,k}(w_{d,k}^{(r)}; \Theta) \Big],$$

for every $d \le D$ and $k \le \nu$, where $\circ$ denotes the element-wise product, and

$$\nabla_{(Z,\theta)} \mathbb{E}_{q(F_n)}[\log p(Y_n \mid F_n)] \approx \frac{1}{R} \sum_{r=1}^{R} \sum_{d=1}^{D} \sum_{k=1}^{\nu} \Big[ \nabla_{F_{d,k}} \log p(Y_n \mid F_n^{(r)}) \circ \nabla_{(Z,\theta)} \psi_{d,k}(w_{d,k}^{(r)}; \Theta) \Big].$$

These unbiased estimates often have low enough variance that a single Monte Carlo sample suffices for the approximation [21], however, we will see in Section 4 that this is not the case with the Wishart process, where a numerical instability renders these estimates useless. Finally, the gradients of the lower bound on the right hand side of Eq. (12) with respect to $\mu_{d,k}$ and $S_{d,k}$ may then be approximated by the unbiased estimator

$$\frac{N}{|\mathcal{B}|} \sum_{n \in \mathcal{B}} \Big[ \nabla_{(\mu_{d,k}, S_{d,k})} \mathbb{E}_{q(F_n)}[\log p(Y_n \mid F_n)] \Big] - \nabla_{(\mu_{d,k}, S_{d,k})} \text{KL}[q(U_{d,k}) \,\|\, p(U_{d,k})], \tag{14}$$

where $\mathcal{B} \subseteq \{(X_n, Y_n) : n \le N\}$ is a minibatch of the datapoints. Likewise, the gradients with respect to $Z$ and $\theta$ may be approximated by

$$\frac{N}{|\mathcal{B}|} \sum_{n \in \mathcal{B}} \Big[ \nabla_{(Z,\theta)} \mathbb{E}_{q(F_n)}[\log p(Y_n \mid F_n)] \Big] - \sum_{d=1}^{D} \sum_{k=1}^{\nu} \nabla_{(Z,\theta)} \text{KL}[q(U_{d,k}) \,\|\, p(U_{d,k})]. \tag{15}$$

With the gradient approximations in Eqs. (14) and (15), gradient ascent may now be carried out with a Robbins–Monro stochastic approximation routine.

We can see that an immediate benefit of taking this black-box variational approach is the ease of switching between the Wishart and inverse Wishart processes, requiring only a switch between the appropriate log conditional likelihood function $\log p(Y_n \mid F_n)$, given by Eq. (5) or Eq. (6), in the subroutine computing Eqs. (14) and (15). This implementation is particularly easy with *GPflow* [16], a Gaussian process toolbox built on Tensorflow, as demonstrated with a code snippet in the Appendix.

Note that choosing $M = N$ and fixing the locations of the inducing points $Z$ at the inputs $X$ results in a Wishart or inverse Wishart process model that captures full temporal dependence among the $N$ measurements. In this case, the evaluations of $\log p(Y_n \mid F_n)$ in Eqs. (5) and (6) have computational complexity and memory requirements with respect to $N$ scaling in $O(N^3)$ and $O(N^2)$, respectively. However, another (equally important) advantage of the inducing point formulation and the variational approach to inference is the ability to reduce this computational burden with respect to $N$, if needed. In particular, by selecting $M \ll N$, we end up with *sparse approximations* to the Gaussian processes. For simplicity, assume that $\nu = D$. In this case, producing a Monte Carlo sample from $q(F_{d,k})$ for the minibatch $\mathcal{B}$ scales in $O(N_b^3 + N_b M^2 + M^3)$ time and $O(N_b^2 + N_b M + M^2)$ space, where $N_b := |\mathcal{B}|$. Producing this for every $d \le D$, $k \le \nu$, together with the computation of $\log p(Y_n \mid F_n)$, results in an overall computation in $O(N_b D^3 + D^2(N_b^3 + N_b M^2 + M^3))$ time and $O(D^2(N_b^2 + N_b M + M^2))$ space. Note that all of these complexities scale linearly with the number of samples $R$ used for the Monte Carlo approximations in Eqs. (14) and (15).

## 4   The additive white noise model

In our initial experiments, we found that the inverse Wishart parameterization successfully moved the parameters into a good region of the state space, whereas the Wishart process failed to move

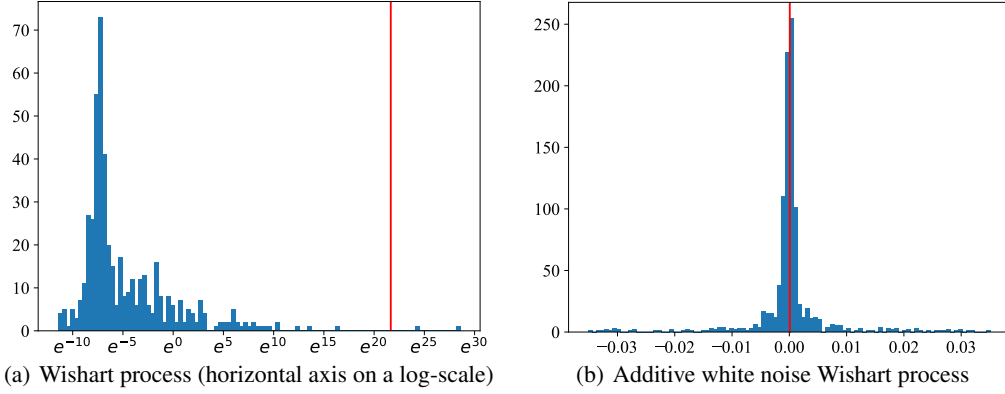

(a) Wishart process (horizontal axis on a log-scale)  (b) Additive white noise Wishart process

Figure 1: Histograms of 1,000 Monte Carlo samples of the gradient with respect to the variable $F_{1,1,1}$ in a univariate model. Fig. 1(a) shows an extremely skewed distribution in the case of the Wishart process, and Fig. 1(b) shows its correction under the additive white noise reparameterization. The mean of each distribution is shown as a red horizontal line.

the parameters in the correct direction (based on traceplots of parameters and validation metrics). It appears that this failure is due to extremely high variance of the Monte Carlo gradient approximation routine. By studying the log-likelihood function for the Wishart process in Eq. (5), we hypothesize that evaluating the inverse in the final term, $-\frac{1}{2}Y_n^T(AF_nF_n^TA^T)^{-1}Y_n$, on Monte Carlo samples of $F_n$ (as required by the procedure described in Section 3) is problematic because those samples can often be close to the origin, resulting in this quantity being extremely large in magnitude. For example, in the case of a univariate output, i.e., $D = 1$, and corresponding unit scale $A = 1$, the likelihood involves computation of the scalar term $-\frac{1}{2}y_n^2/f_n^2$, which is large in magnitude for samples when $f_n$ is closer to zero than the data point $y_n$, a problem that is exacerbated by the quadratic scales.

To visualize this issue, consider a bivariate output $Y_n$ with constant covariance matrix $\Sigma_n = [[2.0, 1.9], [1.9, 2.0]]$ and $A = [[1, 0], [0, 1]]$. We let $\nu = D = 2$ and simulated a dataset $Y_n$ at input locations $X_n$, for $n \leq 30$, which together with some inducing points $Z_m$, $m \leq 10$, are sampled uniformly in $(0, 1)$. As described in Section 3, we compute the following 1,000 samples:

$$\nabla_{F_{d,k}} \log p(Y_n \mid F_n^{(r)}), \quad F_{d,k}^{(r)} \sim q(F_{d,k}), \quad d \leq 2, k \leq 2, n \leq 30, r \leq 1000, \quad (16)$$

and we display a histogram of the samples corresponding to the variable $F_{1,1,1}$ in Fig. 1(a), where the horizontal axis is on a *log scale*. The distribution is extremely skewed; the mean of these samples, at around $2.5 \times 10^9$, is plotted as a red vertical line, and the standard deviation is $7.9 \times 10^{10}$!

To resolve this issue, consider once again the case when $D = 1$ with unit scale $A = 1$. We can modify the previously problematic scalar term to be $-\frac{1}{2}y_n^2/(f_n^2 + \lambda)$, where the denominator is shifted away from zero by a parameter $\lambda > 0$. More generally, we can accomplish this with a slightly generalized construction to that studied by van der Wilk et al. [23]: Construct the covariance matrix of $y_n$ as

$$\Sigma_n := AF_nF_n^TA^T + \Lambda, \quad n \geq 1, \quad (17)$$

where $\Lambda$ is a diagonal $D \times D$ matrix with positive (diagonal) entries. To interpret this modification, note that the model in Section 2 may be alternatively written as $y_n = AF_nz_n$, where $z_n \sim \mathcal{N}(0, I_\nu)$, for $n \geq 1$, so that $\mathrm{Cov}(y_n|F_n) = AF_nF_n^TA^T$. The modified construction may be instead written as

$$y_n = AF_nz_n + \varepsilon_n, \quad z_n \sim \mathcal{N}(0, I_\nu), \quad \varepsilon_n \sim \mathcal{N}(0, \Lambda), \quad n \geq 1, \quad (18)$$

and so $\mathrm{Cov}(y_n|F_n) = AF_nF_n^TA^T + \Lambda$. This modification may therefore be interpreted as introducing *white* (or *observational*) noise to the model. The log conditional likelihood in Eq. (5) is replaced by

$$\log p(Y_n \mid F_n) = \frac{ND}{2}\log(2\pi) - \log|AF_nF_n^TA^T + \Lambda| - \frac{1}{2}Y_n^T(AF_nF_n^TA^T + \Lambda)^{-1}Y_n. \quad (19)$$

The approximated gradients may now be stably computed: In Fig. 1(b), we plot a histogram of the samples of the gradients in Eq. (16) for this modified model, where $\Lambda = [[0.01, 0.0], [0.0, 0.01]]$.

While the inverse Wishart case does not suffer such computational issues, we will see in Section 5 that this additive white noise modification is the key to a *factored* variant of both the Wishart and inverse Wishart processes, inference for which is tractable for high-dimensional covariance matrices. In the inverse Wishart case, however, a useful additive white noise modification is not easy to implement. We consider instead the following construction for the precision matrix

$$\Sigma_n^{-1} := AF_nF_n^TA^T + \Lambda^{-1}, \qquad n \geq 1, \tag{20}$$

where, as a diagonal matrix, $\Lambda^{-1}$ contains the inverted elements on the diagonal of $\Lambda$. If the variables in $y_n$ are independent, then the elements of $\Lambda$ retain their interpretation as (the variances of) additive white noise. More generally, they have an interpretation as additive terms to the *partial variances* of the variables in $y_n$. The log conditional likelihood in Eq. (6) is now replaced by

$$\log p(Y_n \mid F_n) = \frac{ND}{2}\log(2\pi) + \log|AF_nF_n^TA^T + \Lambda^{-1}| - \frac{1}{2}Y_n^T(AF_nF_n^TA^T + \Lambda^{-1})Y_n. \tag{21}$$

The elements of $\Lambda$ share an inverse gamma prior distribution $\Lambda_{d,d}^{-1} \sim \text{gamma}(a,b)$, $d \leq D$, for some $a, b > 0$. We fit a mean-field variational approximation with an analogous approach to the methods in Section 3 for gamma random variables described by Figurnov et al. [5]. (Alternative approaches were described by Knowles [13] and Ruiz et al. [20].) We fit $a$ and $b$ by maximum likelihood.

## 5 Factored covariance models

The computational and memory requirements of inference in the models so far presented scale with respect to $D$ in $O(D^3)$ and $O(D^2)$, respectively, since we must invert (or take the determinant of) a $D \times D$ matrix. This will become intractable for even moderate values of $D$, which is particularly troublesome in applications like finance where $D$ could, for example, represent the number of financial instruments in a large stock market index like the S&P 500. To reduce this complexity, consider fixing some $K \ll D$ and reducing $F_n$ to be of size $K \times \nu$, for some $\nu \geq K$. The matrix $F_nF_n^T$ is a $K \times K$ Wishart-distributed matrix. Let $A$ now be of size $D \times K$. Then by a scaling property of the Wishart distribution [19, p. 535], the $D \times D$ matrix $AF_nF_n^TA^T$ is also Wishart-distributed. This factor-like, low-rank model has significantly fewer parameters than those in Sections 2 and 4.

Consider applying this construction to the additive white noise model for the Wishart process described in Section 4, where $\Sigma_n = AF_nF_n^TA^T + \Lambda$. Recalling that $\Lambda$ is diagonal, the log conditional likelihood function in Eq. (19) may be computed efficiently with the Woodbury matrix identities as

$$\begin{aligned}\log p(Y_n \mid F_n) = \frac{ND}{2}\log(2\pi) - \frac{1}{2}\sum_{d=1}^{D}\log\Lambda_{d,d} + \frac{1}{2}\log|I_\nu + F_n^TA^T\Lambda^{-1}AF_n| \\ -\frac{1}{2}Y_n^T\Lambda^{-1}Y_n + \frac{1}{2}Y_n^TAF_n(I_\nu + F_n^TA^T\Lambda^{-1}AF_n)^{-1}F_n^TA^TY_n.\end{aligned} \tag{22}$$

In the inverse Wishart case, we have $\Sigma_n^{-1} = AF_nF_n^TA^T + \Lambda^{-1}$, which we note is a reparameterization of the construction by Fox and Dunson [6]. The log conditional likelihood function in this case is

$$\begin{aligned}\log p(Y_n \mid F_n) = \frac{ND}{2}\log(2\pi) + \frac{1}{2}\sum_{d=1}^{D}\log\Lambda_{d,d} - \frac{1}{2}\log|I_\nu + F_n^TA^T\Lambda AF_n| \\ -\frac{1}{2}Y_n^T\Lambda^{-1}Y_n - \frac{1}{2}Y_n^TAF_nF_n^TA^TY_n.\end{aligned} \tag{23}$$

For simplicity, assume $\nu = K$. Then these log conditional likelihood functions may be computed (with respect to $D$ and $K$) in $O(DK^2)$ time and $O(DK)$ space. With the black-box approach to variational inference, we need only drop the expressions in Eqs. (22) and (23) into the subroutines computing the gradient estimates in Eqs. (14) and (15). The overall complexity then reduces to computations in $O(N_bDK^2 + K^2(N_b^3 + N_bM^2 + M^3))$ time and $O(DK + K^2(N_b^2 + N_bM + M^2))$ space. This model and inference procedure is therefore scalable to both large $N$ and $D$ regimes.

## 6 Experiments on financial returns

We implement our variational inference routines on the model variants applied to three datasets of financial returns, which are denoted as follows (note that we take the *log returns*, which are defined at time $t + 1$ as $\log(1 + P_{t+1}/P_t)$, where $P_t$ is the price of the instrument at time $t$):

**Dow 30**: Intraday returns on the components of the Dow 30 Industrial Average (as of the changes on Jun. 8, 2009), taken at the close of every five-minute interval from Nov. 17, 2017 through Dec. 6, 2017. The resulting dataset size is $N = 978$, $D = 30$. The raw data was from Marjanovic [14].

**FX**: Daily foreign exchange rates for 20 currency pairs taken from Wu et al. [26]. The dataset size is $N = 1,565$, $D = 20$.

**S&P 500**: Daily returns on the closing prices of the components of the S&P 500 index between Feb. 8, 2013 through Feb. 7, 2018, taken from Nugent [17]. Missing prices are forward-filled. The resulting dataset size is $N = 1,258$, $D = 505$ (there are 505 names in the index).

The simplest baseline is univariate ARCH (applied to each variable independently), implemented through the Python package `arch` [22]. The MGARCH variants we compare to are the *dynamic conditional correlation* model (DCC) [3] with Gaussian and multivariate-t likelihoods, and the *generalized orthogonal garch* model (GO-GARCH) [2], a competitive variant of the BEKK MGARCH specification. These baselines are among the dominant MGARCH modeling approaches and were implemented through the R package `rmgarch` [8]. The MGARCH baselines do not scale to the S&P 500 dataset, and there are no ubiquitous baselines in this large covariance regime.

We used a diagonal matrix $A$ for the full-rank (non-factored) covariance models. The parameters in $A$ and $\Lambda$ are inferred by maximum likelihood. The values of $\Lambda$ were initialized to $\Lambda_{d,d} = 0.001$, $d \leq D$. The degrees of freedom parameter $\nu$ is set to the number of variables $D$, or the number of *factors* $K$ in the factored covariance cases. We did not find performance to be sensitive to this choice. We used $M = 300$ inducing points, $R = 2$ variational samples for the Monte Carlo approximations, and a minibatch size of 300. The gradient ascent step sizes were scheduled according to Adam [11]. We selected the stopping times and an exponential learning rate decay schedule via cross validation, choosing the setting that maximized the test loglikelihood metric (see below) on a validation set. The validation sets were the final 2%, 5%, and 5% of the measurements in just one of the training sets for the Dow 30, FX, and S&P 500 datasets, respectively.

For each dataset, we created 10 evenly-sized training and testing sets with a sliding window, where the test set comprises 10 consecutive measurements following the training set (we may therefore consider a 10-step-ahead forecasting task), and no testing sets overlap. To evaluate the models, we forecast the covariance matrix—say, $\Sigma_t^*$ at horizon $t$, for $t \leq 10$—and compute the log-likelihood of the corresponding test measurement $Y_t$ under a mean-zero Gaussian distribution with covariance $\Sigma_t^*$. The prediction is formed by Monte Carlo estimation with 300 samples from the fitted variational distribution. The parameter settings producing the prediction with the highest training log-likelihood from among a window of 300 steps following the stopping time is kept for testing.

In order to visualize our experimental setup, we display the results for the FX dataset in Fig. 2 as a series of grouped histograms. The horizontal axis represents the forecast horizon; at each horizon, the boxplot of test-loglikelihoods (over the 10 training/testing sets) are displayed for each model. The Wishart and inverse Wishart process variants are denoted by 'wp' and 'iwp', respectively. If the additive white noise parameterization described in Section 4 is used (with a non-factored covariance model), we prepend the model name with 'n-'. The factored model variants, described in Section 5, have model names prepended with 'f[K]-', where [K] is the number of factors. We used a Gaussian process covariance kernel composed as the sum of a Matern 3/2 kernel, a rational quadratic kernel, a radial basis function kernel, and a periodic component, which is itself composed as the product of a periodic kernel and a radial basis function kernel (see the code snippet in the Appendix). The ARCH baseline is denoted 'arch', the DCC baselines with a multivariate normal and multivariate-t likelihood are denoted 'dcc' and 'dcc-t', respectively, and the GO-GARCH baseline is denoted 'go-garch'.

We compare the collections of the log-likelihood scores for each of the 10 forecast horizons in each of the 10 test sets (resulting in populations of 100 scores each). In Table 1, we report the mean score $\pm$ one standard deviation for each model and dataset. For each of our model variants, we provide in brackets the p-value of a Wilcoxon signed-rank test comparing the performance of the model against the highest performing MGARCH baseline (which was always either go-garch or dcc-t) or the ARCH baseline in the case of the S&P 500 dataset. We bold the highest performing model on each dataset, and we highlight any improvements with a $*$ if significant at a 0.05 level.

The Wishart process variants score highest on each dataset; it is notable that they consistently outperform their inverse Wishart process counterparts. In fact, the inverse Wishart process appears to have unreliable performance; while the iwp variants outperform MGARCH on the Dow 30 dataset,

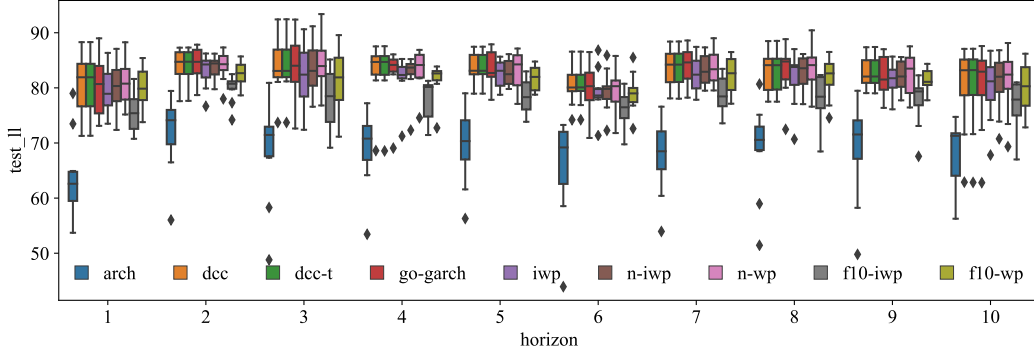

Figure 2: Example display of the results for the FX dataset. A set of boxplots reporting the test loglikelihoods of the predictions is displayed for each step of the 10-step forecast horizon (indicated on the horizontal axis). Each boxplot contains the scores from the 10 training/testing splits.

Table 1: Test loglikelihood metrics across 10-step forecast horizons in 10 test splits. We display the mean over the 100 scores, along with $\pm$ one std. dev. The p-value of a Wilcoxon signed-rank test comparing our models to the highest performing MGARCH/ARCH baseline is displayed in brackets. The highest score is bolded. Significant improvements at a 0.05 level are highlighted with a *.

| | Dow 30 | FX | S&P 500 |
|---|---|---|---|
| arch | $142.47 \pm 17.97$ | $68.24 \pm 7.55$ | $1358.23 \pm 355.12$ |
| dcc | $162.70 \pm 42.98$ | $82.52 \pm 4.55$ | – |
| dcc-t | $162.64 \pm 42.86$ | $82.54 \pm 4.56$ | – |
| go-garch | $163.59 \pm 52.65$ | $82.43 \pm 4.85$ | – |
| iwp | $164.09 \pm 26.47^*$ (1.71e-8) | $81.42 \pm 4.12$ (8.15e-8) | – |
| n-iwp | $164.49 \pm 19.82^*$ (1.13e-9) | $82.10 \pm 3.72$ (1.62e-3) | – |
| n-wp | $\mathbf{165.98 \pm 23.23}^*$ (1.03e-6) | $\mathbf{82.69 \pm 4.15}$ (5.11e-2) | – |
| f10-iwp | $162.28 \pm 22.91^*$ (4.67e-11) | $77.76 \pm 3.94$ (5.99e-17) | $1275.27 \pm 264.48$ (4.14e-18) |
| f10-wp | $165.39 \pm 30.89^*$ (2.31e-5) | $81.12 \pm 3.59$ (2.62e-10) | $\mathbf{1423.31 \pm 132.19}^*$ (1.48e-13) |
| f30-iwp | – | – | $1047.73 \pm 1436.16$ (3.90e-18) |
| f30-wp | – | – | $\mathbf{1438.40 \pm 130.14}^*$ (1.54e-15) |

they perform poorly on the FX and S&P 500 datasets. On the Dow 30 dataset, not only does the (additive noise, full covariance) Wishart process (denoted n-wp) significantly outperform the best performing MGARCH baseline (go-garch, in this case), but every other one of our full covariance models and f10-wp does as well. The f10-iwp model variant is the only one of our models that underperforms go-garch, further emphasizing that the inverse Wishart process should be avoided. While n-wp attains the highest score on the FX dataset, it is not deemed significant over the scores for the highest performing MGARCH baseline (dcc-t in this case), according to the Wilcoxon signed-rank test. However, we may take some comfort in the fact that the p-value of 5.11e-2 (comparing the scores of n-wp and dcc-t) is very close to significance at this level. We note, however, that every other one of our models under-performs compared to dcc-t. The MGARCH baselines cannot scale to the S&P 500 dataset, and so we may only compare our factored covariance models against the diagonal ARCH baseline. The f30-wp and f10-wp models both significantly outperform the ARCH baseline, however, worryingly, the f30-iwp and f10-iwp model variants significantly under-perform the ARCH baseline, giving yet another example of the unreliable performance of the inverse Wishart process.

With this evidence, we recommend a practitioner to always implement the Wishart process instead of the inverse Wishart process. Further study should be undertaken to understand why the performance of these two model variants differ. Unsurprisingly, the additive noise parameterization always improves performance (n-iwp always outperforms iwp), which we may attribute to the additional noise parameters afforded to the model in this case. While we see the factored Wishart process perform competitively with its full covariance counterpart on the Dow 30 dataset, this was the not the case on the FX dataset, and so (unsurprisingly) a full covariance model should be preferred if enough computational resources are available.

# 7 Conclusion

We conclude that the black-box variational approach to inference significantly eases the implementation of the various Wishart and inverse Wishart process models that we have presented. If needed, the computational burden of inference with respect to both the length of the time series and the dimensionality of the covariance matrix may be reduced. We hope that the initial failure of the black-box variational approach in the case of the Wishart process provides a warning to practitioners that these methods cannot always be trusted to work out of the box. We recommend that practitioners always implement the (additive noise) Wishart process instead of the inverse Wishart process. When the dimensionality of the covariance matrix is large, one may use the factored model, however, a full covariance model should be preferred if computational resources will allow it.

## Appendix: Implementation in GPflow

To demonstrate the ease of implementing our methods, we provide the following 25 lines of Python code implementing the inverse Wishart process in GPflow [16] (version 1.3.0):

```python
import numpy as np
import tensorflow as tf
from gpflow import models, likelihoods, kernels, params, transforms, decors

class InvWishartProcessLikelihood(likelihoods.Likelihood):
    def __init__(self, D, R=1):
        super().__init__()
        self.R, self.D = R, D
        self.A_diag = params.Parameter(np.ones(D), transform=transforms.positive)

    @decors.params_as_tensors  # decorator translating TF tensors for GPflow
    def variational_expectations(self, mu, S, Y):
        N, D = tf.shape(Y)
        W = tf.random_normal([self.R, N, tf.shape(mu)[1]])
        F = W * (S ** 0.5) + mu  # samples through which TF automatically differentiates

        # compute the (mean of the) likelihood
        AF = self.A_diag[:, None] * tf.reshape(F, [self.R, N, D, -1])
        yffy = tf.reduce_sum(tf.einsum('jk,ijkl->ijl', Y, AF) ** 2.0, axis=-1)
        chols = tf.cholesky(tf.matmul(AF, AF, transpose_b=True))  # cholesky of precision
        logp = tf.reduce_sum(tf.log(tf.matrix_diag_part(chols)), axis=2) - 0.5 * yffy
        return tf.reduce_mean(logp, axis=0)

class InvWishartProcess(models.svgp.SVGP):
    def __init__(self, X, Y, Z, minibatch_size=None, nu=None):
        D = Y.shape[1]
        nu = D if nu is None else nu  # degrees of freedom

        # create a compositional kernel function
        kern = kernels.Matern32(1) + kernels.RationalQuadratic(1) + kernels.RBF(1) \\
                + kernels.PeriodicKernel(1) * kernels.RBF(1)

        # almost all work is done by SVGP!
        super().__init__(X, Y, Z = Z, kern = kern,  # notation as in the paper
                        likelihood = InvWishartProcessLikelihood(D, R=10),  # 10 MCMC samples
                        num_latent = D * nu,  # number of outputs (multi-output GP)
                        minibatch_size = minibatch_size)
```

GPflow's abstract class `gpflow.models.svgp.SVGP` is designed to automate gradient-based variational inference with (sparse) Gaussian process models. Our only model-specific computation is for the Monte Carlo approximations of the log-likelihood expression in Eq. (6), which is carried out by the method `InvWishartProcessLikelihood.variational_expectations`. The Tensorflow backend automatically differentiates through these expressions to obtain the gradients described in Section 3. Finally, note that a kernel function is being defined from a composition of several simpler kernel functions, demonstrating one of the many utilities of GPflow; this is the particular composition used in our experiments in Section 6.

## Acknowledgements

We thank anonymous reviewers for feedback. All funding for the experiments were personally provided by CH, who does not have an affiliation for this work.

## Footnotes

[1] Alternatively, we may take $A$ to be the (triangular) cholesky factor of a positive definite matrix $AA^T$.

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
