[Reviews · NeurIPS 2019]

Reviewer 1



The paper makes several contributions to the inference of (inverse) Wishart processes. The paper considerers an existing model where the (inverse) D-dimensional Wishart process is constructed by >D independent Gaussian processes and inferred by observing the values at xn of the corresponding correlated variable Y. The first contribution is the scaling of the model to larger datasets N, as this is a well known issue with GPs. For this the authors use a variational inference approximation the Gaussian processes whose parameters they learn by optimizing the ELBO (estimation of noisy gradients required for this). The authors find that regularizing the (inverse) Wishart matrices by a constant pos def matrix (white noise model) helps performance, especially in the case of the Wishart process. Another issue is the computational complexity due to large output dimensions. The authors approach this problem by modeling unregularized Wishart distributed matrices with a lower rank (less GPs), where the rank is a tuning parameters that trades off precision and cost. The authors evaluate on 3 real world datasets, and compare to competitor methods. Experiments show superior performance especially for the inverse regularized Wishart process. I. Originality: The paper combines essentially two existing methods. In that sense the originality is low. However, it appears to me that getting all of those components to work smoothly together is not a trivial task. Also the white noise model and its analysis, and the factored model are original and practically useful extensions. II. Clarity: The clarity of the paper is high in the sense that most details are included in the derivation, but lower on the other hand for the fact that it is so dense. It might have been an easier read if e.g., the details on the gradient computation in section 3 were moved to the supplements, and instead there would be one or two more plots which showcase the approximations done in section 3 and 5, as well as what the regularizer does in 4. In fact, I find the analysis in 4 very nice because it seems to be a recurring issue (e.g., also in optimization, the hessian) where it is not clear if inferring a matrix or its inverse is the better choice. In that regard it would have been interesting to see how a model that predicts Lambda or Lambda^-1 (learned by iwp-n) for every time step would have performed (i.e, a model that predicts Sigma as being the white noise at every step). In section 4, l.133 it is mentioned that the Monte Carlo gradient approximation has high variance, but in section 3 l. 113 the opposite is claimed. It appear to me that both statements correspond to the same approximation, so could the authors please clarify which of them is correct. The experiments show that the model performs well with regard to predicting the covariance Sigma between different time series, i.e., the performance metric is the likelihood of the model for unseen Y. This is nice, but I am unclear about if this is what those models are actually used for. I could imagine that the goal would be to predict the individual time series’ values Y, not in the sense that the Y are a probable draw from the predictive distribution, but in an estimator sense (an estimator for Y). Is there a way to accomplish this with the proposed model? And if yes, how does it perform? Lastly I have a question about the datasets used. I gather that the novel method is proposed because of scaling issues in N and D. However all datasets proposed have N smaller than what a full GP can fairly easily handle. Also the output dimensions do not seem to be that large. Do I misunderstand sth here? I am not sure if the total complexity using a full GP is stated anywhere in the paper. If a full GP is feasible, could the authors please point me to the baseline that used a full GP instead of the variational approximation since this model would be the most interesting to compare to. I would also appreciate a short sentence about wall-clock runtime of these methods. [Post-Rebuttal: Thank you for your rebuttal. I will stay with my initial score.]

Reviewer 2



Post-response update: Thanks for the response. It clarified my concerns about the data size but raised another issue of the method's utility being mostly in financial modelling, which limits the paper's impact in machine learning community. I'm then not increasing my score. ---- The paper applies sparse stochastic variational inference to Wishart process inference, and also proposes low-rank and conditioned wishart for numerical stability. The idea of applying SVI to Wisharts is good, but of little novelty since this work represents combining two well known techniques. The experiments do not highlight the full potential of the method: all datasets are so small that exact inference should be no problem. The method is applied to forecasting of future covariance, but the problem is not motivated. Why is this a meaningful issue to tackle? There is also no discussion of the resulting covariances. With SP500 one gets over 500 dimensional covariance. I don’t see much benefit in modelling these, especially without sparsity. The experiments show only modest improvements. The paper needs to show how the variational inference is necessary with large datasets with thousands to millions of points. The paper also should study how the correlation structures evolve, and why these are useful. The paper is well written, but the math notation is cluttered with lots of subscripts, and no distinction between vectors and matrices with boldface. Given these deficiencies, this paper is not up to nips quality.

Reviewer 3



The author(s) derive a gradient-based variatational inference routine for Wishart and inverse-Wishart processes, which they use to model the dynamic covariance matrices of multi-variate time series. Their motivation is tractability as existing Wishart and inverse-Wishart rely on MCMC methods that require exact posterior inference to be performed on the underlying Gaussian processes. Their approach scales better with time-series length and has a "simpler" implementation. The proposed methods include both variatational inference for Wishart and inverse-Wishart processes as well as their corresponding low-rank factored variants. Their gradient-based method, as they admit, is seemingly largely based on the model from van der Wilk et al. [18]. Their exposition and derivations of their gradient-based variatational inference for Wishart and inverse-Wishart processes are well done despite requiring the reader to keep up with lots of notation. Because their expected log conditional likelihoods cannot be expressed analytically, they use the "reparameterization trick" of Salimans and Knowles [16] and Kingma and Welling [11]. In the case of the Wishart process, they resolve hypothesized numerical instability resulting from matrix inversion, via the introduction of diagonal-covariance Gaussian noise--this adds to the diagonal of the covariance matrix inversion. They confirm via simulation that this proposal resolves numerical instability for the Wishart process. Their factored covariance models use a low-rank approximation to reduce computational complexity from O(D^3) to O(K^2D) for K << D. In the inverse Wishart case, they note their model is a reparemeterization of Fox and Dunson [5]'s construction. Their previously introduced white noise addition allows for efficient matrix inversions via the Woodbury matrix identities. They have three data sets with varied D to which they compare baseline methods to their proposed methods: {Wishart, inverse-Wishart}x{standard, additive noise}x{full, factored} (note factored models required noise). They propose 3 metrics. The metrics being closely related seemingly leads to similar conclusions regrading the relative performances of the algorithms. In the cases of Dow 30 and FX, D is small enough to run all competing baseline methods. There they attain statistically significant results on both tasks with their (inverse-Wishart, additive noise, full). While additive noise was not necessary for numerical instability in the case of the inverse-Wishart, it offers a performance benefit that they hypothesize arises from introducing an additional parameter. For the S&P 500, D was large enough that only the independence-assuming baseline could be used against their factor models. Here, both the baseline method and their proposals have large variance leading their f30-iwp model being best albeit statistically less significant. In summary, this is a well written paper. I will argue it is lacking on the contribution side as the models are all closely related to prior works. That said, their application is well-motivated, and they do achieve competitive results. Further, they do offer solutions for tractability problems for use on real-world data problems. However, since computational efficiency is one of the prime motivations of this paper, it would have been nice to see convergence plotted according to wall-clock time to complement their asymptotic analysis. Their other stand-out finding was the numerical instability associated with the Wishart process, which they both address and warn others to use with caution.

[Author Response · NeurIPS 2019]

We thank the reviewers for their helpful feedback.

**Motivation for variational inference and concern on dataset sizes**

We did not clearly describe the computational challenge when using the (inverse) Wishart process. In particular, all
reviewers state that our dataset sizes are manageable with exact inference; this is incorrect. Exact inference with the
(inverse) Wishart process model is *intractable for even small dataset sizes*. As far as we know, an expression for the
(exact) posterior distribution does not even appear anywhere in the literature. The only options for inference are to set
up an MCMC procedure (as done in previous work) or to take the variational approach we now propose. We made this
lack of clarity worse in the introduction (ll. 30–31) and in Sec. 3 (ll. 81–83) where we discuss "exact posterior inference
on the underlying Gaussian processes." This was unclear wording: the aforementioned MCMC routines require an
exact GP posterior inference computation (costing $O(N^3)$ on each of the $O(D^2)$ GPs) to be performed *during each*
*step of their iterative algorithms*! So we clarify that our variational approach dramatically reduces this *per iteration*
computational cost (while also having an easier implementation via black-box, gradient-based inference and not losing
competitive performance). We will certainly clarify this in the paper.

All reviewers considered the datasets in Sec. 5 to be small, however, note that there are no previous examples in the
literature of inference with the Wishart process scaling to this size (of both $N$ and $D$). We reiterate: our experiments
are the largest scale we're aware of in any work using the (inverse) Wishart process. We would like to point out the
inherently overparameterized nature of this problem: We are estimating a sequence of $N$ covariance matrices, each of
$O(D^2)$ size, from a *single example*. So note that even $D = 20$ is NOT "small". This will always be a difficult task even
for small $N$ and $D$, and the problem will always be overparameterized no matter how big your dataset gets. With that
said, there is no reason to believe that our variational approach would not scale to larger dataset sizes.

Finally, we also note that one additional benefit of the variational approach is a natural way to implement *online*
*inference*. We can also mention this in the paper, though this is certainly not a point we're trying to emphasize.

**Motivation for predicting the covariance matrices**

R1 & R2 requested motivation for why one would want to predict the covariances $\Sigma_n$. The prominent application
requiring such predictions is in the construction of optimal financial trading portfolios that minimize risk (), where
returns are maximized based on a separate model for $Y_n$ (not addressed in this paper) and a model for the *covariance of*
*the residuals* is used to penalize risky (i.e., volatile) assets, which is the part we are tackling in our paper. It is for this
reason that we present in the context of financial applications. Moreover, these predictions of "volatility" or "risk" are
desired throughout finance and beyond. R4 specifically requested motivation beyond finance: such models have been
used to analyze the spread of disease incidence () and XXX (). We will make these clarifications and additions.

R2 similarly questions why (large) covariance matrices would be useful without sparsity. Note that the *full* covariance
matrix is required to construct an optimal financial portfolio. Sparse approximations are often imposed for computational
reasons, but you will *always* construct a suboptimal portfolio with a sparse approximation. This would certainly never
be desired if you can handle the computational burden.

**Originality/novelty**

All reviewers expressed concerns that our work appears too close to previous techniques. We re-emphasize that our work
demonstrates the first alternative to MCMC for inference in the Wishart process model (improving on its computational
efficiency and ease of implementation), and the first experiments scaling inference to the size of the datasets in Sec. 5.

**Other points**

R1 points out the apparent contradiction of "low-variance" MC gradient estimates (through reparameterization) vs. the
study in Sec. 4. We should have instead said: "while such gradient estimates typically have low variance, the particular
form of the Wishart process likelihood introduces computational instability that renders these estimates useless."

R1 & R4 requested some discussion of wall clock runtime of the methods. We will add these to the paper.

We are pleased that R4 recognized the significance of our identification and resolution of the computational issues
involved with applying black-box variational inference to the Wishart process case. These techniques are often applied
indiscriminately in practice, and it is our hope this study (the likes of which are rarely explored in the literature) will
provide a useful warning to practitioners.

We will follow R1's suggestion to move the details of Sec. 3 to the supplement (R2 & R4 also point out the density/clutter
and notation, which we will address); this will also free up enough space to add all the proposed clarifications.

[Meta-Review · NeurIPS 2019]

This paper introduces a novel variational approximation for inference in (inverse) Wishart processes. The reviewers debated this work extensively in the post-rebuttal stage. They were worried about an empirical focus on specific financial modelling applications, but agreed that the paper offers interesting insights into the more general problem of inferring covariance matrices. Overall, I agree with the reviewers that it offers insights of interest to a significant sub-community at NeurIPS, and thus can be accepted to the conference